# Trends in Real-World Clinical Outcomes of Patients with Anaplastic Lymphoma Kinase (ALK) Rearranged Non-Small Cell Lung Cancer (NSCLC) Receiving One or More ALK Tyrosine Kinase Inhibitors (TKIs): A Cohort Study in Ontario, Canada

**DOI:** 10.3390/curroncol32010013

**Published:** 2024-12-27

**Authors:** Lara Chayab, Natasha B. Leighl, Mina Tadrous, Christine M. Warren, William W. L. Wong

**Affiliations:** 1Leslie Dan Faculty of Pharmacy, University of Toronto, Toronto, ON M5S 3M2, Canada; mina.tadrous@utoronto.ca; 2Princess Margaret Cancer Centre, Toronto, ON M5G 2M9, Canada; natasha.leighl@uhn.ca; 3Department of Medicine, University of Toronto, Toronto, ON M5S 3M2, Canada; 4Women’s College Research Institute, Toronto, ON M5G 1N8, Canada; 5ICES, Toronto, ON M4N 3M5, Canada; christine.warren@ices.on.ca; 6School of Pharmacy, University of Waterloo, Waterloo, ON N2L 3G1, Canada; william.wl.wong@uwaterloo.ca

**Keywords:** ALK inhibitors, ALK TKIs, NSCLC, lung cancer, real-world outcomes, alectinib, crizotinib, Ontario Canada, first line treatment, sequential treatment

## Abstract

The treatment landscape for patients with advanced ALK-positive NSCLC has rapidly evolved following the approval of several ALK TKIs in Canada. However, public funding of ALK TKIs is mostly limited to the first line treatment setting. Using linked provincial health administrative databases, we examined real-world outcomes of patients with advanced ALK-positive NSCLC receiving ALK TKIs in Ontario between 1 January 2012 and 31 December 2021. Demographic, clinical characteristics and treatment patterns were summarized using descriptive statistics. Kaplan–Meier analysis was performed to evaluate progression-free survival (PFS) and overall survival (OS) among the treatment groups. A total of 413 patients were identified. Patients were administered alectinib (n = 154), crizotinib (n = 80), or palliative-intent chemotherapy (n = 55) in the first-line treatment. There was a significant difference in first-line PFS between the treatment groups. The median PFS (mPFS) was not reached for alectinib (95% CI, 568 days—not reached), compared to 8.2 months (95% CI, 171–294 days) for crizotinib (HR = 0.34, *p* < 0.0001) and 2.4 months (95% CI, 65–100 days) for chemotherapy (HR = 0.14, *p* < 0.0001). There was no significant difference in first-line OS between the treatment groups. In patients who received more than one line of treatment, there was a significant difference in mOS between patients who received two or more lines of ALK TKIs compared to those who received one line of ALK TKI (mOS = 55 months (95% CI, 400–987 days) and 26 months (95% CI, 1448–2644 days), respectively, HR = 4.64, *p* < 0.0001). This study confirms the effectiveness of ALK TKIs in real-world practice and supports the potential benefit of multiple lines of ALK TKI on overall survival in patients with ALK-positive NSCLC.

## 1. Introduction

Non-Small Cell Lung Cancer (NSCLC) is the most commonly diagnosed cancer in Canada and is associated with high mortality rates [1]. It poses a significant burden for patients, healthcare systems, and society due to its increasing incidence and poor survival outcomes [2]. Each year, approximately 25,000 Canadians are diagnosed with NSCLC, 18,000 succumb to the disease, and the five-year survival rate is 25% [2]. Approximately 3–5% of patients with NSCLC have Anaplastic Lymphoma Kinase (ALK) rearrangements [3,4]. ALK rearrangements are genomic alterations that play a critical role in cancer development and progression. Identifying these mutations through comprehensive genomic profiling is essential for selecting the most effective treatments for patients with lung cancer. The standard treatment for patients with advanced ALK-positive NSCLC involves targeted therapies known as ALK tyrosine kinase inhibitors (TKIs) [5,6,7]. These drugs specifically target the ALK protein, thereby inhibiting the cancer growth and survival pathways [8].

The clinical landscape for patients with advanced ALK-positive NSCLC has evolved significantly with the introduction and approval of several ALK TKIs. In Canada, crizotinib, ceritinib, alectinib, brigatinib and lorlatinib are currently approved by Health Canada for the treatment of patients with ALK-positive NSCLC. However, despite their approval, access to these therapies is currently restricted by funding limitations, which typically cover only first-line treatment options.

Crizotinib, the first ALK TKI, received Health Canada approval for later-line treatment in 2012 and for first-line treatment in 2015, representing a major advancement in targeted therapy [9,10]. Before crizotinib became available, patients were treated with chemotherapy doublet combinations followed by monotherapy. Ceritinib was approved for later-line treatment in 2015 and for first-line treatment in 2017, providing an alternative for patients who either progressed on or were intolerant to crizotinib [11,12,13,14]. Alectinib, noted for its superior efficacy and central nervous system penetration, was approved for later-line treatment in 2016 and for first-line treatment in 2017 [15,16,17,18,19]. Brigatinib, with its effectiveness against various ALK resistance mutations, gained approval for later-line treatment in 2018 and for first-line treatment in 2020 [20,21,22]. Most recently, lorlatinib, a third-generation ALK TKI designed to address resistance to earlier-generation TKIs, was approved for later-line treatment in 2020 and for first-line treatment in 2021 [23,24].

In Canada, cancer treatments are funded by provincial and territorial healthcare systems, each with unique policies for drug approval and reimbursement [25]. After cancer treatments have been approved by Health Canada, the pan-Canadian Oncology Drug Review (pCODR) evaluates their clinical and cost-effectiveness and makes funding recommendations, through the Canadian Agency for Drugs and Technologies in Health (CADTH), to the provinces and territories [26]. Each province and territory then makes its own decision on whether to fund the drug based on these recommendations, considering budget and healthcare priorities [25]. In Ontario, Cancer Care Ontario (CCO) oversees cancer treatment funding and delivery [27]. CCO uses CADTH’s recommendations to develop funding guidelines [27]. CADTH has played a critical role in determining access to ALK TKIs, as the most recent funding algorithm predominantly recommends ALK TKIs to first-line treatment in patients with advanced ALK-positive NSCLC [28].

CADTH funding recommendations issued for each ALK TKI were generally consistent with their approved indications. Crizotinib received its funding recommendation for later-line treatment in 2013 and for first-line treatment in 2016 [29,30]. Ceritinib received its funding recommendation in 2017 for patients who have progressed or are intolerant to crizotinib [31]. Alectinib was recommended for later-line treatment in 2017 and for first-line treatment in 2018 [32,33]. Brigatinib received its funding recommendations for patients untreated with ALK TKIs in 2021 [34]. Lorlatinib, the most recent addition, received recommendations for later-line treatment post chemotherapy and for first-line treatment in 2022 [35].

Despite the positive funding recommendations for each ALK TKI, CADTH consequently published a provincial funding algorithm that predominantly recommended newer ALK TKIs for first-line treatment [28]. Historically, when crizotinib was approved for first-line treatment and ceritinib and alectinib for later lines, patients in Ontario received funding for more than one ALK TKI sequentially. However, as all these ALK TKIs have now been approved for first-line treatment, the CADTH funding algorithm has shifted to limit funding to one ALK TKI for the newer next-generation inhibitors. This shift is influenced by the superior benefit that the next-generation ALK TKIs, such as alectinib, brigatinib, and lorlatinib, offer to patients compared to crizotinib and by the limited clinical trial evidence on the benefit of sequential ALK TKIs following first-line ALK TKI progression. Consequently, patients now receive only one of these newer ALK TKIs in first-line treatment, followed by chemotherapy and immunotherapy. Patients that have access to private coverage or opt to pay out of pocket may end up receiving additional lines of ALK TKIs following first-line treatment. These funding limitations present significant challenges for patients and healthcare providers aiming to manage patients with ALK-positive NSCLC effectively, as the potential benefits of optimal treatment sequencing are not fully realized under current public funding guidelines.

To secure funding for multiple lines of ALK TKIs for patients with advanced ALK-positive NSCLC, additional evidence is needed beyond the current data [28]. Specifically, evidence demonstrating the survival benefits of administering more than one line of ALK TKI therapy and evidence comparing the effectiveness of different ALK TKIs in sequential treatments is of value [28]. In the absence of clinical trial data, real-world evidence becomes a key asset to complement existing evidence, address gaps, and confirm the performance of therapies in practice [36,37,38].

While several publications have assessed the outcomes of ALK TKIs in clinical practice, none have utilized data from Ontario, Canada, at a population level [39,40,41,42,43,44,45,46,47]. Ontario is an ideal location for conducting research. It is the most populated province in Canada, offering a large and diverse patient population, and it is home to a robust and accessible infrastructure for research through its linked health databases, such as those maintained by ICES (formerly the Institute for Clinical Evaluative Sciences). As a result, we conducted a retrospective population-based cohort study using Ontario’s health administrative database to examine the real-world clinical outcomes of patients with advanced ALK-positive NSCLC receiving ALK TKIs in Ontario, Canada. Our study aimed to provide insights into patient characteristics, treatment patterns, and survival outcomes for those receiving one or more ALK TKIs. The purpose is to present evidence to support the funding of ALK TKIs beyond the first-line treatment, ensuring patients have access to effective and well-tolerated treatments throughout the course of their disease.

## 2. Methods

### 2.1. Setting and Design

This was a population-based, retrospective cohort study of residents of Ontario, Canada, aged >/= 18 years who were diagnosed with lung cancer and who received ALK TKI therapy between 1 January 2012 and 31 December 2021. Individuals were followed until death, loss of Ontario Health Insurance Plan (OHIP) eligibility for >6 months or for 3 months post data cut until 31 March 2022. Provincial health insurance in Ontario covers physician and hospital services for all residents [27]. This includes diagnostic procedures and any physician and hospital services needed for patients diagnosed with cancer. Prescriptions for cancer drugs, including chemotherapy, targeted therapy and supportive care medications, are covered primarily under the Ontario Drug Benefit program (ODB) [48]. The ODB provides coverage for eligible residents, including those above 65 years of age, individuals on social assistance and individuals with specific medical conditions [48]. Patients in receipt of ALK TKIs outside of Ontario’s publicly funded health system are not reflected in this study. This may include patients between the ages of 25 and 65 years who are not covered under ODB and who do not qualify to receive social assistance in Ontario [48]. The study was conducted using a prespecified analysis plan approved by the University of Toronto Research Ethics Board (Protocol #: 43068). This study was reported according to the STROBE guideline for cohort studies [49].

### 2.2. Data Sources

Multiple databases across Ontario are linked using unique encoded identifiers and analyzed at the Institute for Clinical Evaluative Sciences (ICES) (Appendix A) [50]. The Ontario Cancer Registry (OCR) database of ICES, which contains up-to-date information on all individuals living in Ontario and diagnosed with cancer, was used to identify individuals diagnosed with lung cancer [51]. Patients receiving ALK TKIs and/or chemotherapy were identified using the Ontario Drug Benefit Program (ODB) database, the Cancer Activity-Level Reporting (ALR) database, and the New Drug Funding Program database, which together cover most of the drugs dispensed in inpatient and outpatient settings [48,52,53]. Data pertaining to hospitalizations and emergency department use were obtained from the Canadian Institute for Health Information Discharge Abstract Database and the National Ambulatory Care Reporting System database [54,55]. The OHIP claims database was used to identify claims for all publicly funded physician services [56]. The date of death was obtained from Ontario Registered Person’s Database (RPDB) [57]. These databases have excellent data completeness and quality, and the collected data are routinely used for research [50,58].

### 2.3. Study Cohort Design

#### 2.3.1. Patients in the Total Study Cohort

We identified a cohort of patients with an Ontario cancer record, who were diagnosed with lung cancer and who were administered a minimum of one ALK TKI treatment in any line of treatment as available in the ODB dataset. The index date of study enrollment was defined as the date of lung cancer diagnosis. The study initiation period was based on the first approval of an ALK TKI in Canada: crizotinib, which was approved in 2012. The study end date was selected based on the most recent time point where ICES had a data update and allowed a 3-month follow-up period at the time of the data cut. Lung cancer diagnosis was defined using ICD codes (See Appendix A). ALK TKIs were defined using the drug identification numbers (DINs) of approved ALK inhibitor treatments in Canada (See Appendix A). There is currently no data field that identifies patients with ALK rearrangements in ICES’s data dictionary. In addition, there are currently no data fields in ICES’s data dictionary that reliably capture information on the administration of genomic testing and/or on the results of genomic tests. Therefore, treatment with ALK TKIs was used as a proxy to identify patients with ALK-positive NSCLC. Patients’ data were retrieved up to 2 years prior to diagnosis and for an additional 3 months after the end of inclusion. Individuals were excluded if they met any of the following criteria: (1) invalid patient ICES key number (IKN); (2) missing age or sex; (3) invalid birth date (missing, after index date or age > 105 or <18 yrs); (4) invalid death date (e.g., before index date); (5) non-Ontario resident at index date; (6) ineligible for OHIP coverage at index date.

#### 2.3.2. Defining Treatments, Treatment Duration, Total Lines of Treatments

Treatment regimens at the individual patient level were retrieved from the ODB and the ALR databases. A lung cancer treatment regimen may include chemotherapy, immunotherapy, ALK TKIs, other targeted therapy or clinical trial therapy. Chemotherapy regimens included different types of chemotherapy treatment(s) and treatment combinations, which were all grouped under the one chemotherapy treatment category for the purposes of this study (see Appendix A).

The treatment duration was derived based on the first treatment administration date, the last treatment administration date and the number of days for which a treatment was supplied. For a patient to be considered to have received a lung cancer treatment regimen, more than one treatment administration date needed to be available to further ensure that patients indeed received the drug. One line of treatment was assumed for patients that received chemotherapy doublet treatment followed by chemotherapy monotherapy treatment.

The total lines of treatment were established at the individual patient level. This is defined as the total number of treatment regimens that an individual received during the study duration. The ALR database contained information on ALK TKIs that were not available in ODB. For the purpose of calculating the total lines of treatment, data on ALK TKIs from the ALR database were also considered and presented as part of the total lines of treatment count. For a complete listing of rules defining the treatments, duration of treatments, and lines of treatment, please refer to Appendix A.

#### 2.3.3. Patients in the Updated Total Study Cohort

After defining treatments, the duration of treatment and the total lines of treatment at the individual patient level, an additional set of patients were excluded from the total study cohort and from further analysis. Patients were further excluded if (1) they received crizotinib in 2018 and beyond, since crizotinib was approved for an additional ROS1 NSCLC indication at that time; and (2) they had only one administration date of treatment in total.

#### 2.3.4. Patients in the First-Line Study Cohort

Patients that were administered a first instance of a cancer treatment regimen, defined as an ALK TKI from ODB or a chemotherapy from ALR, were included in the first-line study cohort. In addition to the exclusions listed in Section 2.3.2 and Section 2.3.3 above, patients were further excluded from this cohort if (1) the first instance of an ALK TKI was retrieved from the ALR database to uphold the inclusion criteria of this study; (2) there was a large gap (>365 days) between the last administration of the first cancer treatment and the next administration of the second cancer treatment, to minimize treatments that may have been intended for adjuvant/neoadjuvant settings; and (3) the first instance of a cancer treatment regimen had only one administration date. Patients were then divided into three treatment groups based on observed ALK TKI administration in the first-line treatment. Patients that were administered alectinib in the first-line treatment were assigned to the alectinib group, patients that were administered crizotinib in the first-line treatment were assigned to the crizotinib group and patients that were administered chemotherapy were assigned to the chemotherapy group. The ODB database did not contain other ALK TKIs administered in the first-line treatment, which is appropriate, since funding recommendations for brigatinib and lorlatinib were issued right at (or after) the end of the study period.

#### 2.3.5. Patients in the First-Line Progressed Study Cohort

Patients in the first-line study cohort who were administered a next line of lung cancer treatment regimen post first-line were included in this cohort. A next line of treatment regimen was defined as an ALK TKI from ODB or ALR, chemotherapy, immunotherapy, other targeted therapies or clinical trials from ALR and with more than one treatment administration date. These patients were considered to have progressed from first-line treatment. Patients were then divided into two groups. The first group of patients were administered two or more ALK TKI agents throughout their treatment journey in any line of treatment and may have also been administered additional lung cancer treatment regimens like chemotherapy and/or immunotherapy in other lines of treatment. The second group of patients were administered one ALK TKI throughout their treatment journey in any line of treatment and would have been administered other lung cancer treatment regimens like chemotherapy and/or immunotherapy in the other lines of treatment.

Figure 1 provides a study flow chart of the total, updated total, first-line and first-line progressed study cohorts.

### 2.4. Outcomes

The baseline characteristics and treatment patterns of the total study cohort were summarized and presented in detailed tables and graphs. Baseline characteristics that were summarized included age, sex, year of diagnosis, income and diversity quintiles. Treatment patterns that were summarized included an overview of the distribution and frequency of ALK TKIs and chemotherapy administrations. This descriptive component aimed to offer insights into the demographic and clinical features of the study population and their treatment experiences.

For the first-line and post-first line progressed study cohorts, baseline characteristics, such as age, sex, year of diagnosis, and total lines of treatment, were assessed between the treatment groups to ensure balance and comparability. Age and sex were evaluated to confirm balance. If the treatment groups were similar in these variables, the study cohorts were considered balanced, and no further adjustments were made.

Year of diagnosis and total lines of treatment were assessed to validate the comparability of the study cohorts. Differences in the year of diagnosis within the first-line study cohort were anticipated, as crizotinib and chemotherapy were the standards of care in earlier years and serve as historical comparators to alectinib. Additionally, differences in total lines of treatment were expected, as patients in the crizotinib and chemotherapy arms had more time to receive additional treatments compared to those in the alectinib arm.

In the post-first line progressed study cohort, differences in the calendar year of diagnosis between the treatment groups were also anticipated. This is because public funding for sequential ALK TKI therapy was primarily available during the time of the earlier generation of ALK TKIs and not recommended for public funding during the later years of the next-generation ALK TKIs, as explained in the introduction section of this manuscript, Section 1.

The primary outcome measured was overall survival (OS), defined as the time from the start of treatment to death or the end of the study period. If death did not occur during the study period, survival was censored at the end of study period. Secondary outcomes included progression-free survival (PFS) and annual survival analysis. PFS was defined as the time from initiation of a first-line treatment regimen until the end of treatment or death from any cause. If treatment discontinuation did not occur during the study timeframe, progression was censored at the end of the study period. Annual survival analysis examined the percentage of patients surviving at the 1-year, 2-year and 3-year time intervals since the start of treatment. No patients were lost to follow-up in the study. All censoring data were reported.

Subgroup analyses were performed to evaluate the impact of variables such as age, sex, year of diagnosis, and total lines of treatment on survival outcomes. These analyses aimed to provide a comprehensive understanding of the additional factors that may have influenced survival in patients receiving ALK TKIs.

### 2.5. Statistical Analysis

Demographic, clinical characteristics and treatment patterns of the study cohorts were summarized using descriptive statistics, and distributions of features between treatment groups in the first-line and post-first line progressed study cohorts were compared using chi square test for categorical variables.

For patients in the first-line study cohort, Kaplan–Meier analysis was performed to evaluate PFS and OS outcomes among the treatment groups. Survival curves were generated for PFS and OS endpoints, and the log-rank test was utilized to compare survival distributions between treatment groups. For patients in the post-first line progressed study cohort, only OS outcomes were compared between the treatment groups. PFS analysis was not conducted, as patients may have received ALK TKIs in any line of treatment setting.

For both cohorts, annual survival analysis involved calculating the percentage of patients surviving at each time interval throughout the study duration.

A subgroup analysis was conducted using the Cox proportional hazards regression model to assess the impact of various covariates on survival outcomes within the treatment groups. The following set of prespecified covariates were included in the analysis: age, sex, year of diagnosis, treatments and total lines of treatment. Proportional hazards assumptions were tested through graphical methods to ensure the model assumptions were met. This was conducted by plotting the Kaplan–Meier survival curve for the different groups and confirming via visual inspection that the curves do not cross (see Appendix A). Hazard ratios (HRs) and corresponding 95% confidence intervals (CIs) were calculated to quantify the association between covariates and survival outcomes.

All of the above analyses were prespecified. All statistical analyses and data manipulations were performed using Stata version 9.4. Statistical significance was defined as *p* < 0.05.

## 3. Results

### 3.1. Total Study Cohort and Updated Total Study Cohort

A total of 413 patients with lung cancer who were treated with ALK TKIs were identified over a period of approximately 10 years (January 2012 to December 2021). The median follow-up time, defined as the time from diagnosis to death or end of follow-up, was 796 days (27 months), with approximately half of the patients (49%) still alive at the time of analysis. The demographic and clinical characteristics of the cohort are summarized in Table 1. The median age was 64 years, with 85% of patients being 50 years or older at the time of lung cancer diagnosis. Females comprised 56% of the cohort. Most patients were diagnosed with lung cancer during the latter half of the study period, with only 23% diagnosed between 2012 and 2015, and the remainder diagnosed post year 2015. Patients generally had a mild Charlson Comorbidity Index rating, and the ratings were distributed across all categories of the Diversity and Income Quintiles.

As described in Section 2.3.3, a total of 41 patients who received crizotinib in 2018 or later, where it may have been intended for the treatment of ROS1 NSCLC, were excluded from further analysis. Additionally, 32 patients who did not receive any cancer treatment regimen were also excluded. After these exclusions, the updated study cohort comprised 340 patients.

The treatment regimens of patients in the updated total study cohort were assessed. The majority (62%) were administered only one line of any cancer treatment regimen, while 38% were administered two or more lines. Among those administered ALK TKIs in any treatment line, 73% received only one ALK TKI, whereas 24% received two or more ALK TKIs. Twenty-three percent of the cohort were administered cytotoxic chemotherapy in any treatment line. Patients administered only one ALK TKI mostly received either alectinib or crizotinib. Those who received two or more ALK TKIs were typically given crizotinib followed by alectinib and/or ceritinib, or alectinib followed by brigatinib and/or lorlatinib. Twenty-three percent of the cohort received cytotoxic chemotherapy in any treatment line.

As shown in Figure 2, in earlier years of diagnosis, a larger percentage of patients were administered multiple lines of treatment, whereas in more recent years, a higher proportion of patients received only one line of therapy. Figure 3 demonstrates that while ALK TKI administration increased over time among lung cancer patients, the use of multiple lines of ALK TKIs treatments, though more common in earlier years, remained consistently low.

### 3.2. Patients in the First-Line Study Cohort

Patients initially received either alectinib (n = 154), crizotinib (n = 80), or palliative-intent cytotoxic chemotherapy (n = 55). The patients’ demographic and clinical characteristics are summarized in Table 2 and mPFS and mOS are depicted in Figure 4 and Figure 5. There were no significant differences in age and sex among the alectinib, crizotinib, and chemotherapy treatment groups. As described in Section 2.4, since age and sex were similar between the groups, the groups were considered to be balanced and no further adjustment to match the groups was conducted.

As anticipated, there was a significant difference in calendar year of diagnosis and in total lines of treatment between the groups. Almost all patients in the alectinib group were diagnosed with lung cancer in more recent years, with 95% diagnosed between 2018 and 2021. In contrast, the majority of patients in the crizotinib and chemotherapy groups were diagnosed between 2012 and 2017 (91% and 64%, respectively, (*p* < 0.001)). Most patients in the alectinib group (87%) received only one line of treatment, whereas the majority of patients in the crizotinib and chemotherapy groups received two or more lines of treatment (56% and 87%, respectively, (*p* < 0.001)). These results confirm the comparability of the groups, as crizotinib and chemotherapy may be indeed used as historical comparators.

Few patients (10%) in the alectinib group went on to receive additional ALK TKIs after their initial alectinib treatment. However, a significant proportion of patients in the crizotinib and chemotherapy groups received one or more additional ALK TKIs following their initial treatment (54% and 85%, respectively).

During a median follow-up of 21 months for alectinib, 45 months for crizotinib, and 26 months for chemotherapy, a significant difference in PFS was observed between the treatment groups. The median PFS (mPFS) was not reached for alectinib (or heavily censored at 30 months, 95% CI, 568 days—not reached), compared to 8.2 months (95% CI, 171–294 days) for crizotinib (HR = 0.34, *p* < 0.0001) and 2.4 months (95% CI, 65–100 days) for chemotherapy (HR = 0.14, *p* < 0.0001). However, there was no significant difference in overall survival (OS) between the groups. The median OS (mOS) was not reached for alectinib (or heavily censored at 39 months, 95% CI, 838 days—not reached), compared to 39 months (95% CI, 454–1430 days) for crizotinib (HR = 0.82, *p* = 0.49) and 31 months (95% CI, 591–1089 days) for chemotherapy (HR = 0.78, *p* = 0.49).

At the end of follow-up, 68% of patients in the alectinib group were still alive, compared to 30% in the crizotinib group and 27% in the chemotherapy group. As shown in Table 3, the percentages of patients surviving at 1-year, 2-year, and 3-year intervals were 83%, 74%, and 69% for the alectinib group; 66%, 55%, and 51% for the crizotinib group; and 80%, 64%, and 42% for the chemotherapy group.

A forest plot of hazard ratios assessing the impact of multiple variables on survival outcomes is presented in Figure 6. There was no significant association between sex, age, and year of diagnosis on survival outcomes. However, treatment with chemotherapy and crizotinib showed a significant negative association with survival when alectinib was used as the reference variable. Additionally, receiving two or more lines of treatment was significantly positively associated with survival compared to receiving only one line of treatment.

### 3.3. Patients in the Post-First Line Progressed Cohort

Of patients who received additional lung cancer treatments post first-line, n = 73 received two or more ALK TKIs, while n = 39 received only one ALK TKI. As presented in Table 4, there were no significant differences in age, sex, and total lines of treatment between the groups. As described in Section 2.4, since age and sex were similar between the groups, the groups were considered balanced and no further adjustment to match the groups was conducted. As expected, patients in the two or more ALK TKI group were diagnosed with lung cancer earlier than those in the one ALK TKI group (*p* = 0.01).

Patients in the one ALK TKI group received either crizotinib or alectinib in any treatment line, along with other regimens like chemotherapy and/or immunotherapy in other treatment lines. For patients in the two or more ALK TKI group, Table 4 provides a breakdown of the ALK TKIs administered. About half of patients in the two or more ALK TKI group were administered crizotinib followed by alectinib, while the other ~half of patients were administered other ALK TKIs such as alectinib followed by lorlatinib or brigatinib, and a few received three or more ALK TKIs in total.

At the end of follow-up, 49% of patients in the two or more ALK TKI group were still alive, compared to 31% in the one ALK TKI group. As shown in Figure 7, there was a significant difference in mOS between patients who received two or more ALK TKIs compared to those who received one ALK TKI (mOS = 55 months (95% CI, 400–987 days) and 26 months (95% CI, 1448–2644 days), respectively, HR = 4.64, *p* < 0.0001). Progression-free survival was not compared in the post first-line progressed study cohort due to the different lines of treatment that the ALK TKIs were administered and the small sample size, which made it impractical to compare each line of treatment post first-line.

## 4. Discussion

To our knowledge, this is the first study to examine the real-world treatment patterns and outcomes of patients with lung cancer who received one or more ALK TKI therapies in any line of treatment at a population level in Ontario, Canada. Our analysis of a cohort of 413 patients covered a 10-year period (January 2012–December 2021) with a median follow-up of approximately 27 months, capturing baseline characteristics, treatment patterns, total lines of treatment, and outcomes.

### 4.1. Total Cohort

The median age of the total study cohort was 64 years, which is older compared to the median age of patients with ALK-positive NSCLC reported in previous studies at ~55 years of age [15,16,17,18,19,59,60,61]. This discrepancy is likely due to the underrepresentation of patients aged 25 to 65 who are not covered by the Ontario Drug Benefit (ODB) program [48]. The majority of patients were diagnosed with lung cancer during the latter half of the study period, potentially reflecting an increased uptake of genomic testing for driver mutations in Ontario over time [62]. However, due to limited data availability in ICES regarding the administration of genomic testing, trends in the uptake of such testing could not be confirmed in this study. Consequently, 49% of patients were still alive at the time of analysis.

Patients diagnosed earlier in the study period received multiple lines of treatment, while those diagnosed more recently typically received only one line. This trend likely reflects the availability of more effective treatments such as ALK TKIs, as well as newer and more effective ALK TKIs like alectinib compared to crizotinib over time. Notably, patients who received only one line of ALK TKI treatment were mostly administered alectinib, whereas those who received multiple lines were typically treated with crizotinib followed by alectinib and/or ceritinib. This aligns with the CADTH reimbursement review, which recommended funding alectinib and/or ceritinib after progression on first-line crizotinib but did not recommend funding any other ALK TKIs post first-line alectinib progression [28].

### 4.2. First-Line Study Cohort

An analysis of patients in the first-line study cohort confirms the evolving standard of care over time. In this study, chemotherapy was predominantly administered as first-line treatment for patients diagnosed with lung cancer until 2015. This shifted to crizotinib between 2015 and 2018, and more recently, alectinib became the predominant first-line treatment from 2019 to 2021. The progression of first-line treatment observed in this study aligns with the approval and funding of ALK TKIs in Ontario, Canada [29,30,31,32,33]. While newer-generation ALK TKIs, such as brigatinib and lorlatinib, are now funded for first-line treatment, their administration was not captured due to the study’s end date being slightly earlier than the date that funding decisions were made for these drugs [34,35]. Interestingly, the study showed some first-line cytotoxic chemotherapy administration even in recent years, possibly indicating delays in testing and/or access to treatments.

Because patients in the alectinib group were diagnosed more recently, their progression and survival outcomes were mostly censored. Despite the immature data, the mPFS was significantly longer for the alectinib group compared to the crizotinib and chemotherapy groups (mPFS not reached for alectinib (or heavily censored at 30 months), 8.2 months for crizotinib, and 2.4 months for chemotherapy, HR = 0.4, *p* < 0.0001). On the other hand, there was no significant difference in mOS between the groups (mOS not reached for alectinib (or heavily censored mOS of 39 months), 39 months for crizotinib, and 31 months for chemotherapy, HR = 0.4, *p* = 0.5). However, the 1-year, 2-year, and 3-year survival rates were higher in the alectinib-treated cohort, with a 3-year survival rate of 69% for alectinib, 51% for crizotinib, and 42% for chemotherapy. Additionally, a Cox proportional hazards regression showed a statistically significant negative association with crizotinib and chemotherapy on survival and a significant positive association between patients who received two or more lines of cancer treatments and survival. In this study, the log-rank test did not show a significant difference in survival distributions between treatment groups, but the Cox proportional hazards regression test showed a significant association within the same treatment groups on survival. This may indicate that the log-rank may not have been sensitive enough to detect differences typically due to the sample size and the number of events impacting the power of the test.

The results from the first-line study cohort align with published clinical trial and real-world evidence comparing the effectiveness of alectinib to crizotinib and chemotherapy [19,47,63,64,65,66]. In the PROFILE 1014 trial, which compared crizotinib with chemotherapy as first-line treatment for advanced ALK-positive NSCLC, the mPFS was significantly longer in patients who received crizotinib (10.9 months) compared to chemotherapy (7.0 months; HR, 0.45, 95% CI, 0.35–0.60; *p* < 0.001) [10]. The median duration of treatment was 10.9 months in the crizotinib group versus 4.1 months in the chemotherapy group [10]. The mOS was not reached for crizotinib (95% CI, 45.8 months to NR) and was 47.5 months for chemotherapy, with 82.5% of the chemotherapy arm crossing over to crizotinib (95% CI, 32.2 months to NR) [10]. The 4-year survival probability was 56.6% (95% CI, 48.3–64.1%) with crizotinib and 49.1% (95% CI, 40.5–57.1%) with chemotherapy [10].

In the ALEX trial, which compared alectinib to crizotinib as first-line treatments in patients with advanced ALK-positive NSCLC, the mPFS was 34.8 months for alectinib versus 10.9 months for crizotinib (HR = 0.43, 95% CI 0.32–0.58) [63]. The median treatment duration was longer with alectinib at 28.1 months compared to 10.8 months for crizotinib [63]. The mOS was not reached with alectinib, whereas it was 57.4 months with crizotinib (stratified HR 0.67, 95% CI 0.46–0.98) [63]. Additionally, the 5-year overall survival rate was 62.5% (95% CI 54.3–70.8) for alectinib versus 45.5% (95% CI 33.6–57.4) for crizotinib [63].

Although the mOS in this study is slightly lower than observed in clinical trials across all three treatment cohorts, this may result from the differences in patient selection between clinical trials and real-world practice. Furthermore, the older median age of patients in this study compared to those in clinical trials could be another contributing factor to the shorter mOS observed. Other real-world data studies report similar outcomes, with mOS being slightly shorter in practice compared to clinical trial data. For instance, Noronha et al. (2016) found that patients treated with first-line crizotinib in the real world achieved an mPFS of 10 months and an mOS of 39.8 months [64]. Similarly, Reynolds et al. (2018) reported an mPFS of 10 months and an mOS of 33.8 months for patients treated with first-line crizotinib in the real world [65]. In a more recent study by Gibson et al. (2021), exploring real-world clinical outcomes of patients with ALK-rearranged advanced NSCLC treated with alectinib or crizotinib in Alberta, Canada, the mPFS was 16 months for crizotinib and 34.9 months for alectinib, while the mOS was 46.6 months for crizotinib and 50.7 months for alectinib [47].

These findings from previous real-world studies and our study re-affirm effectiveness of ALK TKIs in practice and further confirm the superior benefit of alectinib over crizotinib and over chemotherapy in the first-line treatment setting, supporting the shift to alectinib as the standard of care for patients with advanced ALK-positive NSCLC.

### 4.3. Post First-Line Progressed Study Cohort

An analysis of patients in the post first-line progressed study cohort revealed a significantly longer median overall survival (mOS) in patients who received two or more ALK TKIs compared to those who received only one ALK TKI throughout their cancer journey (55 months vs. 26 months, HR = 4.64, *p* < 0.0001). Approximately half of the patients in the two or more ALK TKI group received an older sequencing regimen (crizotinib followed by alectinib) that is no longer the standard of care. However, the other half were treated with next-generation ALK TKIs (such as alectinib, brigatinib, and/or lorlatinib), underscoring the relevance of these findings to current Canadian treatment practices. It is also worth noting that the significant difference in mOS observed between the treatment groups in the post first-line progressed study cohort may explain the lack of significant mOS observed between the treatment groups in the first-line study cohort. This is because most patients in the first-line crizotinib and first-line chemotherapy treatment groups were administered additional ALK TKIs in later lines of treatment, unlike the alectinib treatment group.

The results from this study are consistent with findings from other studies, which showed that administering two or more ALK TKIs to patients with advanced ALK-positive NSCLC results in an mOS of up to 59 months, whereas administration of one ALK TKI is associated with a lower mOS ranging from 16 to 41 months [40,41,42,43,44,45,46,66,67,68,69].

In ElSayed et al. (2021), mOS was compared between patients with advanced ALK-positive NSCLC who received two or more ALK TKIs and those who received one ALK TKI with chemotherapy or one ALK TKI with no further treatment [66]. The study showed an mOS of 59 months (95% CI 43–74) for patients with two or more ALK TKIs, 41 months (95% CI 26–55) for patients with one ALK TKI and chemotherapy (log-rank *p* = 0.002), and 16 months (95% CI 8–23) for patients with one ALK TKI only (log-rank *p* < 0.001) [66]. Similarly, Britschgi et al. (2020) found that the use of more than one ALK TKI positively correlated with overall survival (*p* = 0.016), with patients treated with more than one ALK TKI achieving an mOS of 85.7 months (95% CI: 63.9–107.5) compared to 34.8 months (95% CI: 21.6–48.0) for patients who received one ALK TKI [67].

Unfortunately, these real-world results have not been confirmed in larger clinical trials of sequential ALK TKI administration. Most clinical trials conducted for alectinib, brigatinib, and lorlatinib in the second-line treatment setting post-progression on crizotinib were single-arm studies [15,20,23,70]. The ALUR and ALTA-3 trials, which included control arms, provide some evidence of the benefits of sequential ALK TKIs [71,72,73]. In the ALUR study, alectinib was compared to chemotherapy in the third-line treatment setting post-crizotinib and platinum-based chemotherapy, with significantly longer mPFS for alectinib (10.9 vs. 1.4 months; HR 0.20, 95% CI 0.12–0.33; *p* < 0.001) [74]. The ALTA-3 trial compared brigatinib to alectinib in the second-line setting post-progression on crizotinib, showing no significant difference between the two ALK TKIs, but highlighting the potential benefit of sequential ALK TKI therapy with an mPFS of 19.3 months for brigatinib and 19.2 months for alectinib, and mOS was not reached for either treatment arm [73].

Although the benefit of sequential ALK TKIs has yet to be confirmed in clinical trials, multiple real-world data studies, including this study, demonstrate the potential benefit of sequential ALK TKI treatments compared to only one line of ALK TKI treatment for patients with advanced ALK-positive NSCLC.

### 4.4. Limitations

Given the retrospective nature of this study, the survival analysis had several limitations. The analysis was based on non-randomized patient cohorts. Although the groups were similar in age, sex, and total lines of treatment, the lack of balance in years of lung cancer diagnosis resulted in more mature data for the crizotinib and chemotherapy groups compared to the alectinib group in the first-line study cohort and a smaller sample size of patients initially treated with alectinib in the post-first line progressed study cohort. Additionally, due to database limitations, important baseline characteristics such as performance status, smoking status, and brain metastasis, which are known to influence survival in patients with ALK-positive NSCLC, were not included in any of the analyses.

Although the study data from the ICES database were complete, many patients were excluded from the study cohorts. The exclusions were due to either patients having one administration of a cancer treatment regimen, which did not meet the study’s definition of an administered treatment, or having a large gap between treatment regimens, potentially indicating treatment prior to advanced disease, which could not be confirmed. It is also important to note that genomic rearrangements are not consistently and reliably reported in the ICES database. As a result, the study cohort was identified using the diagnosis of lung cancer and treatment with ALK TKIs to identify patients with advanced ALK-positive NSCLC. While this method is generally reliable, as it is unlikely that ALK TKIs would be administered without an ALK rearrangement, it limited the study to patients who received ALK TKI treatment and excluded those with ALK-positive NSCLC who might have adhered to other treatment regimens.

In addition, due to public funding restrictions on oral cancer medications in Ontario, the ODB dataset was enriched for individuals above the age of 65 and those who are eligible for Ontario’s public drug plans [48]. Furthermore, ICD codes used for lung cancer diagnosis may have included lung cancer patients with other subtypes such as small-cell lung cancer (SCLC). As it is unlikely that patients with other subtypes were prescribed ALK TKIs, as their use is not indicated in other subtypes, our data may still have included other subtypes of lung cancer. In addition, while we based our inclusion criteria on ALK TKI administration data from the ODB database, we identified additional ALK TKIs in the ALR database that were not reflected in the ODB. This discrepancy excluded some patients from the study cohort that may have been eligible if the inclusion criteria had been broader to incorporate all patients with lung cancer receiving ALK TKIs from any database.

Moreover, patients that were administered different types of chemotherapy regimens were all grouped together into one overarching chemotherapy treatment group. Although this approach enabled a larger sample size to be available for a chemotherapy treatment group to be represented in this study, it assumed homogeneity within the group, potentially leading to skewed or generalized findings that do not reflect the outcomes associated with any of the specific chemotherapy regimens.

Lastly, assignment of patients in the post-first line progressed treatment groups were based solely on the total number of ALK TKI treatments administered, without considering the line of treatment setting (e.g., first-line, second-line) or the specific type of ALK TKI therapy. Although this approach allows for a sufficient sample size in each treatment group for comparison, it also introduces significant variability within groups. For example, patients at different stages of their treatment journey may have differing responses, disease progression rates, and prior treatment influences. Consequently, the results from the post-first line progressed study cohort help provide an overarching view of potentially better survival outcomes in patients that were administered two or more ALK TKIs throughout their cancer journey but should not be used to compare the effectiveness of specific sequential ALK TKI regimens, as the sequencing and timing of specific therapies was not accounted for in this cohort.

Despite these limitations, this study was able to examine the effectiveness of ALK TKIs in practice in the first-line treatment setting and generate outcomes of patients based on the total number of ALK TKIs administered. This study also confirms the ability of the ICES database to be a reliable source at a population level for examining the outcomes of patients diagnosed with lung cancer and administered ALK TKIs in Ontario due to its strong connectivity to multiple data sources and the completeness of the data variables within it.

### 4.5. Future Research and Impact on Current Policy

This study supports previous findings from clinical trials and from real-world studies, demonstrating that patients diagnosed with lung cancer who were treated with alectinib had significantly longer mPFS compared to those treated with crizotinib or chemotherapy in the first-line setting. Additionally, our results confirm that the benefit of sequential treatment with ALK TKIs observed in other real-world studies also applies to patients in Ontario. Specifically, patients who received two or more ALK TKIs had significantly longer mOS compared to those who received only one ALK TKI and other cancer regimens.

Despite alectinib being established as the first-line standard of care (with brigatinib and lorlatinib recently added as options), there is currently no public funding available in Canada for ALK TKIs beyond the first-line treatment. The evidence to date supporting the benefits of sequential ALK TKI therapy suggests that this issue cannot be easily overlooked. Patients in Ontario who progress after first-line treatment with alectinib, brigatinib, or lorlatinib may either access additional ALK TKIs through private insurance or pay out of pocket. This situation presents a moral dilemma for clinicians, who must navigate access restrictions while advising patients on suitable subsequent therapies, particularly those with limited financial means. From a payer’s perspective, the absence of confirmatory clinical trials on sequential ALK TKI treatment complicates the decision to fund additional ALK TKIs beyond the first line.

Based on our study’s findings, we recommend periodic updates to assess the effectiveness of ALK TKIs with more mature data, including for the alectinib cohort, and to evaluate new evidence for brigatinib and lorlatinib in the first-line setting. Brigatinib demonstrated superior efficacy in the ALTA-1L study as a first-line treatment for ALK-positive NSCLC, achieving an mPFS of 24.0 months compared to 11.0 months with crizotinib (hazard ratio [HR]: 0.49, 95% CI: 0.35–0.68), along with a favorable safety profile and enhanced intracranial activity [22]. Lorlatinib showed remarkable efficacy in the CROWN study, achieving an mPFS of 33.3 months compared to 9.3 months with crizotinib (hazard ratio [HR]: 0.28, 95% CI: 0.19–0.41), highlighting its significant advantage in delaying disease progression in patients with ALK-positive NSCLC, while maintaining a manageable safety profile [24]. We also recommend periodic updates to enable larger sample size for the post-first line progressed cohort that may enable the assignment of patients into treatment groups based on the number of ALK TKIs administered, the line of treatment of ALK TKI administration and the specific type of ALK TKI administered. Moreover, we recommend broadening the inclusion criteria to ALK TKIs from multiple ICES data sources to better capture sequential ALK TKI therapy. Additionally, we suggest conducting a network meta-analysis of existing real-world studies on sequential ALK TKI treatments. This analysis could provide a solid basis for submitting updated recommendations to CADTH for extended funding of ALK TKIs beyond the first line.

Lastly, although genomic testing was out of scope in this particular study, it is worth emphasizing that the treatment of patients with ALK-positive NSCLC is closely tied to the ability to comprehensively test for genomic alterations. Timely and appropriate identification of genomic mutations at the time of diagnosis, including specific fusion partners, co-occurring mutations and resistance mechanisms at the time of relapse, can guide optimal selection of initial and sequential ALK TKI therapies for better patient outcomes [74]. In Ontario, Canada, the detection of ALK rearrangements in patients with NSCLC primarily utilizes immunohistochemistry (IHC) as the initial screening method, followed by confirmatory testing with more specific methods like FISH or next-generation sequencing [75]. IHC is quick and cost-effective, but it does not provide the detailed molecular information needed to identify fusion partners. Next-generation sequencing (NGS) is more frequently being integrated into clinical practice for comprehensive genomic profiling, capable of detecting a wide array of genetic alterations simultaneously, including ALK rearrangements and their specific fusion partners [76]. However, the adoption of NGS in routine clinical settings has been gradual, influenced by factors such as cost, access, availability of infrastructure, and the need for specialized expertise [76]. To address these challenges and enhance the accessibility of comprehensive biomarker testing, Ontario has implemented the Comprehensive Cancer Biomarker Testing Program [77]. This initiative aims to standardize and coordinate biomarker testing across the province, ensuring that patients have access to evidence-based and comprehensive testing. While the program has enhanced the adoption of molecular testing for NSCLC in Ontario, the routine integration of comprehensive genomic profiling, both at diagnosis and upon progression in patients with ALK-positive NSCLC, is still evolving and varies across different healthcare settings [78,79]. In addition, while NGS is widely regarded as the most effective method for detecting genomic alterations, including specific fusion partners, concerns remain about NGS potentially missing certain fusions [80]. This can arise from factors such as incomplete panel coverage, insufficient RNA quality or quantity, degraded samples from formalin-fixed tissues, and limitations in bioinformatics pipelines that may fail to detect novel or complex fusion variants [80].

Ensuring that all patients have access to both the necessary testing and targeted treatments is crucial. Future research should not only explore trends in treatment patterns and outcomes for patients with ALK-positive NSCLC in real-world settings, but should also focus on addressing the technical and logistical challenges that hinder the widespread adoption of comprehensive genomic profiling. Overcoming these barriers will be critical to ensuring that every patient has the opportunity to access the testing and treatments required for the best possible care.

## 5. Conclusions

In summary, this study is the first to examine real-world treatment patterns and outcomes for lung cancer patients in Ontario who received one or more ALK TKI therapies across any line of treatment at a population level. Our findings indicate that treatment patterns for ALK TKIs align with CADTH’s funding recommendations. Initially, patients received chemotherapy when it was the standard of care, transitioned to crizotinib during its standard-of-care period and, more recently, adopted alectinib as the new standard.

Consistent with public funding practices, the study reveals that most patients who started with chemotherapy or crizotinib later received additional ALK TKIs, whereas those who began treatment with alectinib rarely received subsequent ALK TKIs. While the effectiveness of ALK TKIs has been validated in clinical trials, this study underscores their significant impact in real-world settings as well. Specifically, the analysis of first-line treatments showed that patients treated with alectinib had a significantly longer median progression-free survival compared to those treated with crizotinib or chemotherapy. Additionally, among patients who progressed from first-line treatment, those who received two or more ALK TKIs had a significantly longer median overall survival compared to those who received only one ALK TKI.

Overall, this study confirms the effectiveness of ALK TKIs in real-world practice and provides further evidence supporting the potential benefit of multiple lines of ALK TKI therapy on overall survival for patients with advanced ALK-positive NSCLC. 

## Figures and Tables

**Figure 1 curroncol-32-00013-f001:**
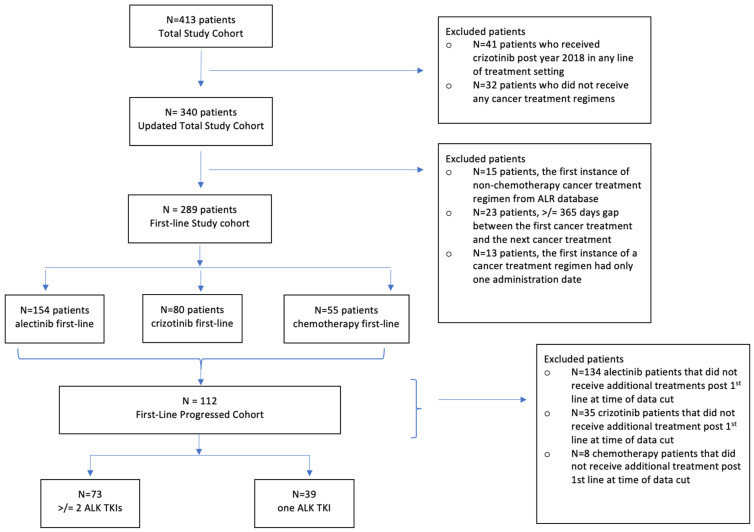
Study flow chart.

**Figure 2 curroncol-32-00013-f002:**
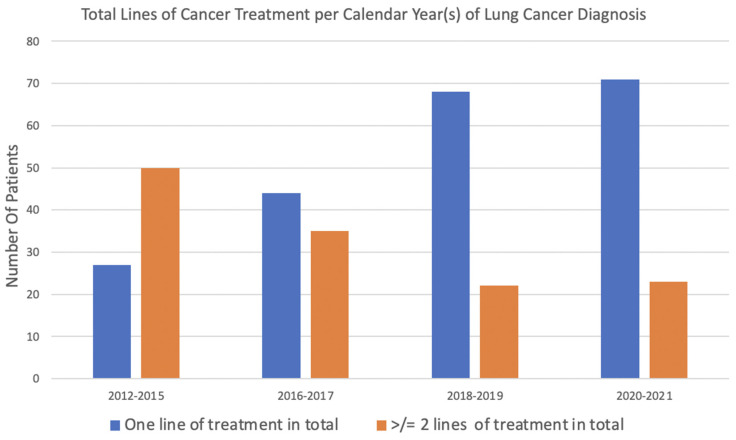
Lines of treatment of any cancer treatment regimen per calendar year(s) of diagnosis.

**Figure 3 curroncol-32-00013-f003:**
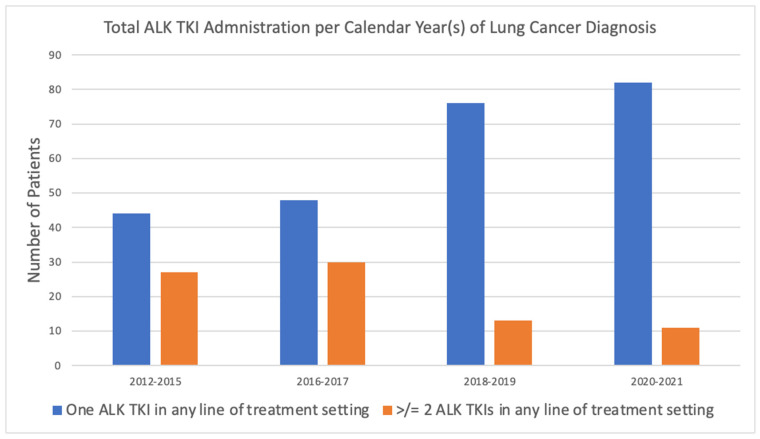
Total of ALK TKIs administered in any line of treatment setting per calendar year(s) of diagnosis.

**Figure 4 curroncol-32-00013-f004:**
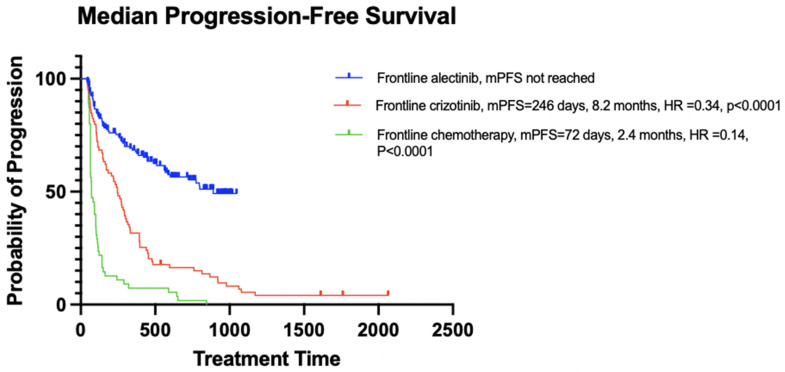
PFS of patients in the first-line treatment cohort.

**Figure 5 curroncol-32-00013-f005:**
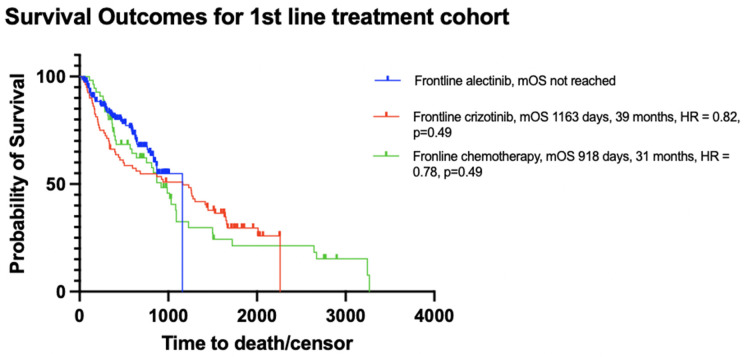
OS of patients in the first-line treatment cohort.

**Figure 6 curroncol-32-00013-f006:**
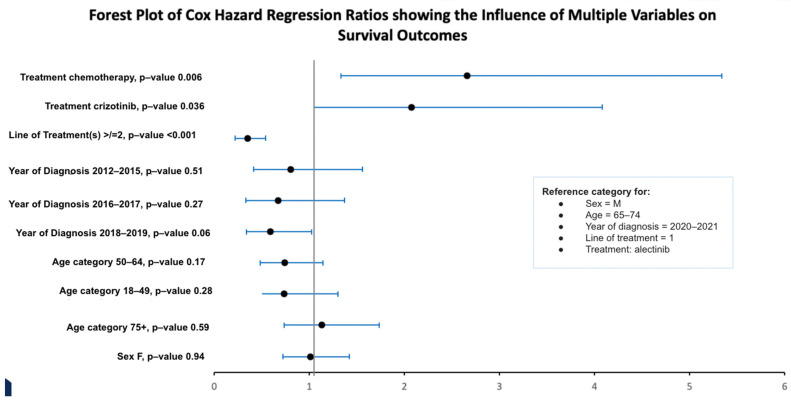
Forest plot assessing the impact of multiple variables on survival outcomes using the Cox proportional hazards model.

**Figure 7 curroncol-32-00013-f007:**
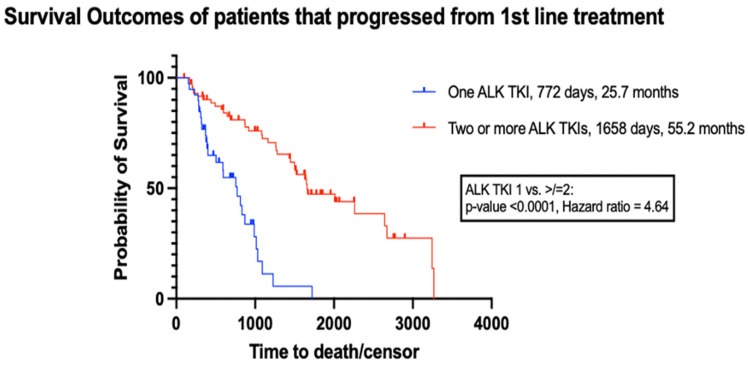
Survival outcomes of patients that progressed from first-line treatment and received one or more ALK TKI agents in any line of treatment setting.

**Table 1 curroncol-32-00013-t001:** Demographics and clinical features of total cohort.

Variable	Total Cohort (n = 413)
Age at Diagnosis	
Mean age of cohort at Diagnosis of Lung Cancer (years)	64
Age	
Age category 18–49 yrs n (%)	60 (15)
Age category 50–64 yrs n (%)	136 (33)
Age category 65–74 yrs n (%)	135 (32)
Age category 75+ n (%)	82 (20)
Sex	
Female n (%)	231 (56)
Calendar year of Diagnosis	
2012–2015 n (%)	95 (23)
2016–2017 n (%)	90 (22)
2018–2019 n (%)	102 (25)
2020–2021 n (%)	126 (30)
Number of patients alive at end of follow-up	
Alive n (%)	204 (49)
Death n (%)	209 (51)
Days from Diagnosis to Death/end date	
Median follow-up in days (months)	796 (27 months)
Charlson Index	
0–1; mild n (%)	397 (96)
2–10; moderate to severe n (%)	16 (4)
Diversity Quintiles	
1—low n (%)	59 (14)
2 n (%)	43 (11)
3 n (%)	56 (14)
4 n (%)	97 (23)
5—high n (%)	157 (38)
Income Quintile	
1—low n (%)	92 (22)
2 n (%)	71 (17)
3 n (%)	88 (21)
4 n (%)	75 (18)
5—high n (%)	87 (21)
Patients excluded from Total Study Cohort	
Patients who received crizotinib post year 2018 in any line of treatment setting (n) Period where crizotinib ALK indication and public funding overlapped with crizotinib ROS1 indication and public funding Excluded from further analysis	41
Patients who did not receive any cancer treatment regimen (n) Patients with less than one ‘days to serve/visit’ date for a cancer treatment regimen	32
Lines of Treatment	
Total patients in all lines of treatment (n) Excluding n = 41 patients who received crizotinib post 2018 and n = 32 patients who received no cancer treatment regimen	340
Total patients administered one line of treatment n (% of 340 patients)	210 (62)
Total patients administered 2 lines of treatment n (% of 340 patients)	81 (24)
Total patients administered >/= 3 lines of treatment n (% of 340 patients)	49 (14)
ALK TKI administration	
Total patients administered ALK TKIs n (% of 340 patients)	331 (97) n = 184 administered alectinib n = 63 administered crizotinib
Number of patients administered one ALK TKI in any line of treatment setting n (% of 340 patients)	250 (73) n = 52 administered crizotinib followed by alectinib and/or ceritinib n = 14 administered alectinib followed by brigatinib and/or lorlatinib n = 15 other
Number of patients administered 2 or more ALK TKIs in any line of treatment setting n (% of 340 patients)	81 (24)
Total patients who were not administered any ALK TKIs	9 (3) These were patients that met inclusion criteria of administration of a minimum of one ALK TKI treatment in any line of treatment setting as available in the ODB dataset; however, ALK TKI administration had one ‘days to serve/visit’ date.
Number of patients who were administered cytotoxic chemotherapy	
Chemotherapy n (% of 340 patients)	77 (23)
No chemotherapy n (% of 340 patients)	263 (77)

**Table 2 curroncol-32-00013-t002:** Demographics and clinical features of first-line treatment cohort.

Variable (First-Line Treatment Setting)	Alectinib (n = 154)	Crizotinib (n = 80)	Chemotherapy (n = 55)	*p*-Value
Age at Diagnosis				
Age category 18–49 yrs n (%)	21 (14)	10 (13)	6 (11)	
Age category 50–64 yrs n (%)	50 (32)	23 (29)	16 (29)	
Age category 65–74 yrs n (%)	55 (36)	28 (35)	25 (45)	
Age category 75+ n (%)	28 (18)	19 (23)	8 (15)	
				0.79 ^1^
Sex				
Female n (%)	84 (55)	47 (59)	25 (45)	
				0.31 ^2^
Calendar year of Diagnosis				
2012–2015 n (%)	NR (Not Reported) ^3^	22 (28)	35 (64)	
2016–2017 n (%)	NR ^3^	50 (63)	NR ^3^	
2018–2019 n (%)	66 (43)	NR ^3^	NR ^3^	
2020–2021 n (%)	80 (52)	NR ^3^	11 (20)	
				0.001 ^4,^*
Total Lines of Treatment				
1 n (%)	134 (87)	35 (44)	7 (13)	
>/=2 n (%)	20 (13)	45 (56)	48 (87)	
				0.001 ^5,^*
Number of patients who received one or more ALK TKIs post first-line treatment ^6^	15 (10)	43 (54)	47 (85)	
Number of patients who received chemotherapy post first-line treatment ^7^				
Chemotherapy n (%)	NR ^3^	NR ^3^	NR ^3^	
Number of patients alive at end of follow-up				
Alive n (%)	105 (68)	24 (30)	15 (27)	
Death n (%)	49 (32)	56 (70)	40 (73)	
Number of patients censored				
# of patients achieved progressed PFS	63 (40)	75 (94)	55 (100)	
Days from index date to death/last date of follow-up				
Median follow-up days (months)	647 (21 months)	1358 (45 months)	775 (26 months)	

This table reports the number and proportion of individuals in variables of interest. (1) A Chi2 test was performed between Age and Treatment. All expected cell frequencies were greater than 5; thus, the assumptions for the Chi2 test were met. There was no statistically significant relationship between Age and Treatment, χ^2^(6) = 3.13, *p* = 0.793, Cramér’s V = 0.07. (2) A Chi2 test was performed between Sex and Treatment. All expected cell frequencies were greater than 5; thus, the assumptions for the Chi2 test were met. There was no statistically significant relationship between Sex and Treatment, χ^2^(2) = 2.36, *p* = 0.307, Cramér’s V = 0.09. (3) Data suppressed to comply with ICES privacy policies because calculations required the use of a cell involving 5 or fewer patients or might have exposed another cell that contained 5 or fewer patients. (4) * A Chi2 test was performed between Year of Diagnosis and Treatment. All expected cell frequencies were greater than 5; thus, the assumptions for the Chi2 test were met. There was a statistically significant relationship between Year of Diagnosis and Treatment, χ^2^(6) = 249.15, *p* = <0.001, Cramér’s V = 0.66. (5) * A Chi2 test was performed between Total Lines of Treatment and Treatment. All expected cell frequencies were greater than 5; thus, the assumptions for the Chi2 test were met. There was a statistically significant relationship between Total Lines of Treatment and Treatment, χ^2^(2) = 107.58, *p* = <0.001, Cramér’s V = 0.61. (6) In addition to ALK inhibitors post first-line treatment, patients may have received additional lines of treatments like chemotherapy or immunotherapy. (7) In addition to chemotherapy post first-line treatment setting, patients may have received additional lines of treatments like other ALK inhibitors or immunotherapy.

**Table 3 curroncol-32-00013-t003:** % of patients in the first-line treatment cohort surviving at each time interval.

Treatment	1-Year Survival Rate%	2-Year Survival Rate%	3-Year Survival Rate%
Alectinib	83%	74%	69%
Crizotinib	66%	55%	51%
Chemotherapy	80%	64%	42%

**Table 4 curroncol-32-00013-t004:** Demographics and clinical features of post first-line progressed patient cohort.

Variable (Post First-Line Progressed)	>/=2 ALK TKIs (n = 73)	One ALK TKI (n = 39)	*p*-Value
Age at Diagnosis			
Age category 18–49 years n (%)	8 (11)	NR ^1^	
Age category 50–64 years n (%)	24 (33)	13 (33)	
Age category 65–74 years n (%)	28 (38)	14 (36)	
Age category 75+ n (%)	13 (18)	NR ^1^	
			0.72 ^2^
Sex			
Female n (%)	35 (48)	20 (51)	
			0.74 ^3^
Calendar year of Diagnosis			
2012–2015 n (%)	23 (32)	17 (44)	
2016–2017 n (%)	27 (37)	NR ^1^	
2018–2019 n (%)	12 (16)	NR ^1^	
2020–2021 n (%)	11 (15)	12 (31)	
			0.01 ^4,^*
Total Lines of Treatment			
2 n (%)	42 (58)	27 (69)	
>/=3 n (%)	31 (42)	12 (31)	
			0.23 ^5^
Types of ALK TKI agents received			
Alectinib only n (%)	NR ^1^	19 (49)	
Crizotinib only n (%)	NR ^1^	20 (51)	
2 ALK TKI agents in any line of treatment: alectinib, crizotinib n (%)	38 (52)	NA ^1^	
2 ALK TKI agents in any line of treatment other than alectinib, crizotinib ^6^ n (%)	20 (27)		
3 or more ALK TKI agents in any line of treatment setting ^7^ n (%)	15 (21)	NR ^1^	
Number of patients alive at end of follow-up			
Alive n (%)	36 (49)	12 (31)	
Death n (%)	37 (51)	27 (69)	
Days from index date to death/last date of follow-up			
Median follow-up in days (months)	1484 days (49 months)	603 (20 months)	

This table reports the number and proportion of individuals in variables of interest. (1) Data suppressed to comply with ICES privacy policies because calculations required the use of a cell involving 5 or fewer patients or may have exposed another cell that contained 5 or fewer patients. (2) A Chi2 test was performed between Age and Treatment. All expected cell frequencies were greater than 5; thus, the assumptions for the Chi2 test were met. There was no statistically significant relationship between Age and Treatment, χ^2^(3) = 1.36, *p* = 0.716, Cramér’s V = 0.11. (3) A Chi2 test was performed between Treatment and Sex. All expected cell frequencies were greater than 5; thus, the assumptions for the Chi2 test were met. There was no statistically significant relationship between Treatment and Sex, χ^2^(1) = 0.11, *p* = 0.736, Cramér’s V = 0.03. (4) * A Chi2 test was performed between Treatment and Year of Diagnosis. All expected cell frequencies were greater than 5; thus, the assumptions for the Chi2 test were met. There was a statistically significant relationship between Treatment and Year of Diagnosis, χ^2^(3) = 10.67, *p* = 0.014, Cramér’s V = 0.31. (5) A Chi2 test was performed between Treatment and Total Lines of Treatment. All expected cell frequencies were greater than 5; thus, the assumptions for the Chi2 test were met. There was no statistically significant relationship between Treatment and Total Lines of Treatment, χ^2^(1) = 1.47, *p* = 0.225, Cramér’s V = 0.11. (6) Patients received either (alectinib, lorlatinib), (alectinib, brigatinib), (crizotinib, ceritinib) or (crizotinib, lorlatinib) in any line of treatment. (7) Patients received either (alectinib, crizotinib, brigatinib), (crizotinib, alectinib, lorlatinib), (crizotinib, ceritinib, alectinib) or (crizotinib, alectinib, brigatinib, lorlatinib) in any line of treatment.

## Data Availability

The dataset from this study is held securely in coded form at the Institute for Clinical Evaluative Sciences (ICES). While data-sharing agreements prohibit ICES from making the dataset publicly available, access may be granted to those who meet prespecified criteria for confidential access, available at www.ices.on.ca/DAS, accessed on 18 August 2023. The full dataset creation plan and underlying analytic code are available from the authors upon request, understanding that the computer programs may rely upon coding templates or macros that are unique to ICES and are therefore either inaccessible or may require modification.

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
