# Peer review of "Trends in Real-World Clinical Outcomes of Patients with Anaplastic Lymphoma Kinase (ALK) Rearranged Non-Small Cell Lung Cancer (NSCLC) Receiving One or More ALK Tyrosine Kinase Inhibitors (TKIs): A Cohort Study in Ontario, Canada"

_curroncol, 2024, doi:10.3390/curroncol32010013_

Round 1
Reviewer 1 Report
Comments and Suggestions for Authors
The authors reported a real-world experience of patients with ALK positive NSCLC.
The results agree with what was reported in literature about the benefit of multiple lines of ALK TKI therapy on OS.
I have the following suggestions:
- emphasize the importance of comprehensive genomic tumor profiling at the time of diagnosis or of the relapse to guide the TKI selection;
- adding data on Lorlatinib, a third-generation TKI, recently become available as a first-line TKI option for advanced ALK-positive NSCLC, in view of the impressive outcomes showed by the CROWN study.
Author Response
For research article: curroncol-3364151
Dear Reviewer 1,
Thank you very much for taking the time to review this manuscript. Please find the detailed responses attached and the corresponding revisions highlighted in the revised manuscript.
Regards,
Lara Chayab on behalf of all authors

Reviewer 2 Report
Comments and Suggestions for Authors
The manuscript is timely relevant and technically correct requiring minor commnents to be accepted for the publication on this journal
- In the introduction section, please, could the authors forefront the clinically available landscape of Canadian NSCLC ?
- In the discussion section, please, could the authors stress the technical and logistic barriers for the diffusion of molecular testing in real world series?
- In the result section, please, could the author report fusion partner for each ALK alteration?
Author Response
Dear Reviewer 2,
Thank you very much for taking the time to review this manuscript. Please find the detailed responses attached and the corresponding revisions highlighted in the revised manuscript.
Regards,
Lara Chayab on behalf of all authors
